# LIPSCHITZ REGULARIZED DEEP NEURAL NETWORKS GENERALIZE

## ABSTRACT

We show that if the usual training loss is augmented by a Lipschitz regularization term, then the networks generalize. We prove generalization by first establishing a stronger convergence result, along with a rate of convergence. A second result resolves a question posed in Zhang et al. (2016): how can a model distinguish between the case of clean labels, and randomized labels? Our answer is that Lipschitz regularization using the Lipschitz constant of the clean data makes this distinction. In this case, the model learns a different function which we hypothesize correctly fails to learn the dirty labels.

## 1 INTRODUCTION

While deep neural networks networks (DNNs) give more accurate predictions than other machine learning methods (LeCun et al., 2015), they lack some of the performance guarantees of these other methods. One step towards performance guarantees for DNNs is a proof of generalization with a rate. In this paper, we present such a result, for Lipschitz regularized DNNs. In fact, we prove a stronger convergence result from which generalization follows.

We also consider the following problem, inspired by (Zhang et al., 2016).

*Problem* 1.1. [Learning from dirty data] Suppose we are given a labelled data set, which has Lipschitz constant $\text{Lip}(\mathcal{D}) = \mathcal{O}(1)$ (see (3) below). Consider making copies of 10 percent of the data, adding a vector of norm $\epsilon$ to the perturbed data points, and changing the label of the perturbed points. Call the new, *dirty*, data set $\tilde{\mathcal{D}}$. The dirty data has $\text{Lip}(\tilde{\mathcal{D}}) = \mathcal{O}(1/\epsilon)$. However, if we compute the histogram of the pairwise Lipschitz constants, the distribution of the values on the right hand side of (3), are mostly below $\text{Lip}(\mathcal{D})$ with a small fraction of the values being $\mathcal{O}(1/\epsilon)$, since the duplicated images are $\epsilon$ apart but with different labels. Thus we can solve (1) with $L_0$ estimate using the prevalent smaller values, which is an accurate estimate of the clean data Lipschitz constant. The solution of (1) using such a value is illustrated on the right of Figure 1. Compare to the Tychonoff regularized solution on the right of Figure 2. We hypothesis that on dirty data the solution of (1) replaces the *thin tall spikes with short fat spikes* leading to better approximation of the original clean data.

In Figure 1 we illustrate the solution of (1) (with $L_0 = 0$), using synthetic one dimensional data. In this case, the labels $\{-1, 0, 1\}$ are embedded naturally into $Y = \mathbb{R}$, and $\lambda = 0.1$. Notice that the solution matches the labels exactly on a subset of the data. In the second part of the figure, we show a solution with dirty labels which introduce a large Lipschitz constant, in this case, the solution reduces the Lipschitz constant, thereby correcting the errors.

Learning from dirty labels is studied in §2.4. We show that the model learns a different function than the dirty label function. We conjecture, based on synthetic examples, that it learns a better approximation to the clean labels.

We begin by establishing notation. Consider the classification problem to fix ideas, although our restuls apply to other problems as well.

**Definition 1.2.** Let $\mathcal{D}_n = x_1, \ldots, x_n$ be a sequence of *i.i.d.* random variables sampled from the probability distribution $\rho$. The data $x_i$ are in $X = [0, 1]^d$. Consider the classification problem with $D$ labels, and represent the labels by vertices of the probability simplex, $Y \subset \mathbb{R}^D$. Write $y_i = u_0(x_i)$ for the map from data to labels.

Write $u(x; w)$ for the map from the input to data to the last layer of the network.[1] Augment the training loss with Lipschitz regularization

$$\min_{u:X \to Y} J^n[u] = \frac{1}{n} \sum_{i=1}^{n} \ell(u(x_i; w), y_i) + \lambda \max(\text{Lip}(u) - L_0, 0) \tag{1}$$

The first term in (1) is the usual average training loss. The second term in (1) the Lipschitz regularization term: the excess Lipschitz constant of the map $u$, compared to the constant $L_0$.

In order to apply the generalization theorem, we need to take $L_0 \geq \text{Lip}(u_0)$, the Lipschitz constant of the data on the whole data manifold. In practice, $\text{Lip}(u_0)$ can be estimated by the Lipschitz constant of the empirical data. The definition of the Lipschitz constants for functions and data, as well as the implementation details are presented in §1.3 below.

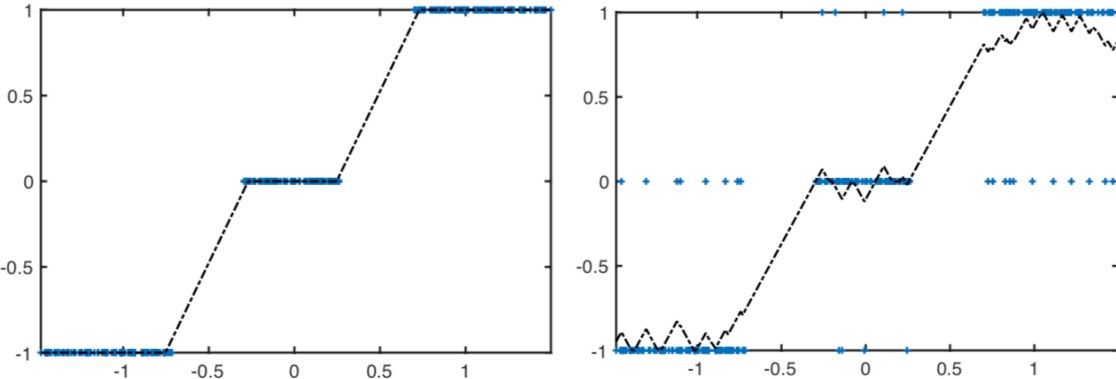

Figure 1: Synthetic labelled data and Lipschitz regularized solution $u$. Left: The solution value matches the labels exactly on a large portion of the data set. Right: dirtly labels: 10% of the data is incorrect; the regularized solution corrects the errors.

Our analysis will apply to the problem (1) which is *convex* in $u$, and does not depend explicitly on the weights, $w$. Of course, once $u$ is restricted to a fixed neural network architecture, the corresponding minimization problem becomes non-convex in the weights. Our analysis can avoid the dependence on the weights because we make the assumption that there are enough parameters so that $u$ can exactly fit the training data. The assumption is justified by Zhang et al. (2016). As we send $n \to \infty$ for convergence, we require that the network also grow, in order to continue to satisfy this assumption. Our results apply to other non-parametric methods in this regime.

## 1.1 RELATED WORK AND APPLICATIONS

Generalization bounds have been obtained previously via VC dimension analysis of neural networks (Bartlett, 1997). The generalization rates have factors of the form $A^k$ for a $k$-layer neural network with bounds $\|w_i\| \leq A$ for all weight vectors $w_i$ in the network. Such bounds are only applicable for low-complexity networks. Other works have considered connections between generalization and stability (Bousquet & Elisseeff, 2002; Xu & Mannor, 2012). More recently, (Bartlett et al., 2017) proposed the Lipschitz constant of the network as a candidate measure for the Rademacher complexity, which is a measure of generalization (Shalev-Shwartz & Ben-David, 2014, Chapter 26). Also, Cranko et al. (2018) showed that Lipschitz regularization can be viewed as a special case of distributional robustness. Unlike other recent contributions such as (Hardt et al., 2015), our analysis does not depend on the training method. In fact, our analysis has more in common with inverse problems in image processing, such as Total Variation denoising and inpainting (Bertalmio et al., 2000; Rudin et al., 1992). For further discussion, see Appendix C.

---

[1]We apologize for not using the standard notation $f$ for the last layer!

The estimate of $\mathrm{Lip}(u; X)$ provided by (4) can be quite different from the the Tychonoff gradient regularization (Drucker & Le Cun, 1992),

$$\frac{1}{|I|} \sum_{i \in I} \|\nabla_x u(x_i)\|^2$$

since (4) corresponds to a maximum of the values of the norms, and the previous equation corresponds to the mean-squared values. In fact, recent work on semi-supervised learning suggests that higher $p$-norms of the gradient are needed for generalization when the data manifold is not well approximated by the data (El Alaoui et al., 2016; Calder, 2017; Kyng et al., 2015; Slepcev & Thorpe, 2017). In Figure 2 we compare to the problems in Figure 1 using Tychonoff regularization. The Tychonoff regularization is less effective at correcting errors. The effect is more pronounced in higher dimensions.

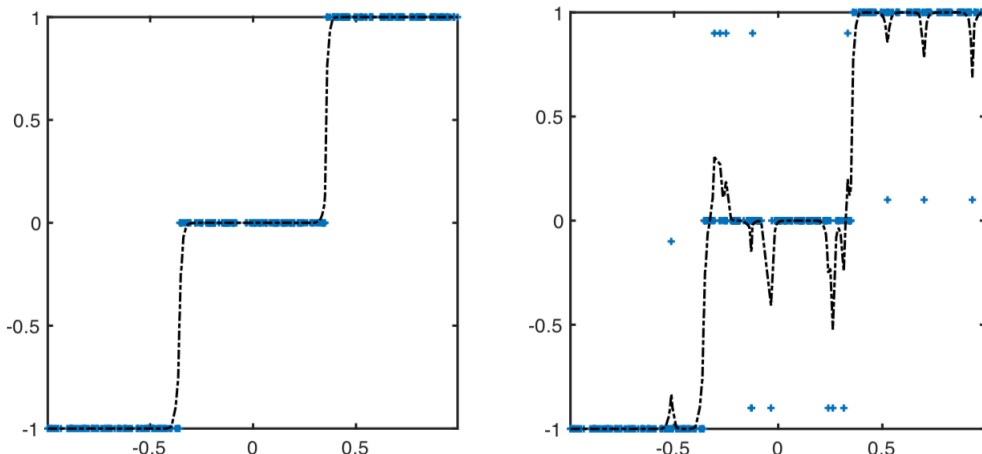

Figure 2: Synthetic labelled data and Tychonoff regularized solution $u$. Left: The solution value matches the labels exactly on a large portion of the data set. Right dirty labels: 10% of the data is incorrect; the regularized solution is not as effective at correcting errors. The effect is more pronounced in higher dimensions.

## 1.2 RELATED WORK ON LIPSCHITZ REGULARIZATION

An upper bound for the Lipschitz constant of the model is given by the norm of the product of the weight matrices (Szegedy et al., 2013, Section 4.3). Let $w = (w^1, \ldots, w^J)$ be the weight matrices for each layer. Then

$$\mathrm{Lip}(u; X) \leq \Pi_{j=1}^{J} \|w^i\|. \tag{2}$$

Regularization of the network using methods based on (2) has been implemented recently in (Gouk et al., 2018) and (Yoshida & Miyato, 2017). Because the upper bound in (2) does not take into account the coefficients in weight matrices which are zero due to the activation functions, the gap in the inequality can be off by factors of many orders of magnitude for deep networks (Finlay & Oberman, 2018).

Implementing (4) can be accomplished using backpropagation in the $x$ variable on each label, which can become costly for $D$ large. Special architectures could also be used to implement Lipschitz regularization, for example, on a restricted architecture, Liao et al. (2018) renormalized the weight matrices of each layer to be norm 1.

Lipschitz regularization may help with adversarial examples (Szegedy et al., 2013) (Goodfellow et al., 2014) which poses a problem for model reliability (Goodfellow et al., 2018). Since the Lipschitz constant $L_\ell$ of the loss, $\ell$, controls the norm of a perturbation

$$\|\ell(u(x_i + \epsilon v)) - \ell(u(x_i))\|_Y \leq \epsilon L_\ell \|v\|_X$$

maps with smaller Lipschitz constants may be more robust to adversarial examples. Finlay & Oberman (2018) implemented Lipschitz regularization of the loss, and achieved better robustness against adversarial examples, compared to adversarial training (Goodfellow et al., 2014) alone.

Lipschitz regularization may also improve stability of GANs. 1-Lipschitz networks are also important for Wasserstein-GANs (Arjovsky et al., 2017) (Arjovsky & Bottou, 2017). In (Wei et al., 2018) the gradient penalty away from norm 1 is implemented, augmented by a penalty around perturbed points, with the goal of improved stability. Spectral regularization for GANs was implemented in (Miyato et al., 2018).

### 1.3 LIPSCHITZ CONSTANTS AND IMPLEMENTATION

**Definition 1.3** (Lipschitz constants of functions and data)**.** Choose norms $\| \cdot \|_Y$, and $\| \cdot \|_X$ on $X$ and $Y$, respectively. The Lipschitz constant (in these norms) of a function $u : X_0 \subset X \to Y$ is given by

$$\text{Lip}(u; X_0) = \sup_{x_1, x_2 \in X_0} \frac{\|u(x_1) - u(x_2)\|_Y}{\|x_1 - x_2\|_X}$$

When $X_0$ is all of $X$, we write $\text{Lip}(u; X) = \text{Lip}(u)$. The Lipschitz constant of the data is given by

$$\text{Lip}(u_0; \mathcal{D}_n) = \max_{x_1, x_2 \in \mathcal{D}_n} \frac{\|u_0(x_1) - u_0(x_2)\|_Y}{\|x_1 - x_2\|_X} \tag{3}$$

Finlay & Oberman (2018) implement Lipschitz regularization as follows. The basis for the implementation of the Lipschitz constant is Rademacher's Theorem (Evans, 2018, §3.1), which states that if a function $g(x)$ is Lipschitz continuous then it is differentiable almost everywhere and $\text{Lip}(g) = \max_x \|\nabla g(x)\|$.

Restricting to a mini-batch, we obtain the following method for estimating the Lipschitz constant. Let $u(x; w)$ be a Lipschitz continuous function. Then

$$\max_{i \in I} \|\nabla_x u(x_i; w)\| \leq \text{Lip}(u; X) \tag{4}$$

For vector valued functions, the appropriate matrix norm must be used, see §B.

## 2 LIPSCHITZ REGULARIZATION AND CONVERGENCE

### 2.1 LIMITING PROBLEM

The variational problem (1) admits Lipschitz continuous minimizers, but in general the minimizers are not unique. When $L_0 = \text{Lip}(u_0)$, it is clear that $u_0$, is a solution of (1): both the loss term and the regularization term are zero when applied to $u_0$. In addition, any $L_0$-Lipschitz extension of $u_0|_{\mathcal{D}_n}$ is also a minimizer of (1), so solutions are not unique.

Let $u_n$ be any solution of the Lipschitz regularized variational problem (1). We study the limit of $u_n$ as $n \to \infty$. Since the empirical probability measures $\rho_n$ converge to the data distribution $\rho$, the continuum variational problem corresponding to (1) is

$$\min_{u:X \to Y} J[u] \equiv L[u; \rho] + \lambda \max(\text{Lip}(u) - L_0, 0), \tag{5}$$

where in (5) we have introduced the following notation.

**Definition 2.1.** Given the loss function, $\ell$, a map $u : X \to Y$, and a probability measure, $\mu$, supported on $X$, define

$$L[u, \mu] = \mathbb{E}_{x \sim \mu}[\ell(u(x), u_0(x))] = \int_X \ell(u(x), u_0(x)) d\mu(x)$$

to be the expectation of the loss with respect to the measure. In particular, the *generalization loss* of the map $u : X \to Y$ is given by $L[u, \rho]$. Write $L[u, \mathcal{D}_n] := L[u, \rho_n]$ for the average loss on the data set $\mathcal{D}_n$, where $\rho_n := \frac{1}{n} \sum \delta_{x_i}$ is the empirical measure corresponding to $\mathcal{D}_n$.

*Remark* 2.2. Generalization is defined in (Goodfellow et al., 2016, Section 5.2) as the expected value of the loss function on a new input sampled from the data distribution. As defined, the full generalization error includes the training data, but it is of measure zero, so removing it does not change the value.

## 2.2 LOSS FUNCTION ASSUMPTIONS

We introduce the following assumption on the loss function.

**Assumption 2.3** (Loss function). The function $\ell : Y \times Y \to \mathbb{R}$ is a *loss* function if it satisfies (i) $\ell \geq 0$, (ii) $\ell(y_1, y_2) = 0$ if and only if $y_1 = y_2$, and (iii) $\ell$ is strictly convex in $y_1$.

*Example* 2.4 ($\mathbb{R}^D$ with $L^2$ loss). Set $Y = \mathbb{R}^D$, and let each label be a basis vector. Set $\ell(y_1, y_2) = \|y_1 - y_2\|_2^2$ to be the $L^2$ loss.

*Example* 2.5 (Classification). In classification, the output of the network is a probability vector on the labels. Thus $Y = \Delta_D$, the $D$-dimensional probability simplex, and each label is mapped to a basis vector. The cross-entropy loss $\ell^{KL}(y, z) = -\sum_{i=1}^D z_i \log(y_i/z_i)$. For labels, $\ell^{KL}(y, e_k) = -\log(y_k)$.

*Example* 2.6 (Regularized cross-entropy). In the classification setting, it is often the case that the softmax function

$$\text{softmax}(z)_j = \frac{e^{z_j}}{\sum_{k=1}^D e^{z_k}} \tag{6}$$

is combined with the cross-entropy loss. In this paper, we regard softmax as the last layer of the DNN, so we assume the output $u(x)$ of the network lies in the probability simplex. If the output, $z$, of the second to last layer of the DNN, which is the input to softmax in (6), lies in a compact set, i.e., $|z_j| \leq C$ for all $i$ and some $C > 0$, then $\text{softmax}(z)_j \geq e^{-2C}$, and so the range of softmax lies in the set

$$A := \{y \in \mathbb{R}^D \; : \; y_i \geq e^{-2C} \text{ and } y_1 + \cdots + y_D = 1\},$$

which is strictly interior to the probability simplex. Restricted to $A$, the cross-entropy loss $\ell^{KL}$ is strongly convex and Lipschitz continuous, which is required in Theorems 2.12 and 2.11 below.

In our analysis, it is slightly more convenient to define the *regularized cross entropy loss* with parameter $\epsilon > 0$

$$\ell_\epsilon^{KL}(y, z) = -\sum_{i=1}^D (z_i + \epsilon) \log\left(\frac{y_i + \epsilon}{z_i + \epsilon}\right).$$

For classification problems, where $z = e_k$, we have $\ell_\epsilon^{KL}(y, e_k) = -(1 + \epsilon) \log((y_k + \epsilon)/(1 + \epsilon))$, which is Lipschitz and strongly convex for any $0 \leq y_i \leq 1$ within the probability simplex. Thus, the regularized cross entropy $\ell_\epsilon^{KL}$ satisfies the strong convexity and Lipschitz regularity required by Theorems 2.12 and 2.11 on the whole probability simplex.

## 2.3 GENERALIZATION RESULT

Here, we show that solutions of the random variational problem (1) converge to solutions of (5). We make the standard manifold assumption (Chapelle et al., 2006), and assume the data distribution $\rho$ is a probability density supported on a compact, smooth, $m$-dimensional manifold $\mathcal{M}$ embedded in $X = [0, 1]^d$, where $m \ll d$. We denote the probability density again by $\rho : \mathcal{M} \to [0, \infty)$. Hence, the data $\mathcal{D}_n$ is a sequence $x_1, \ldots, x_n$ of *i.i.d.* random variables on $\mathcal{M}$ with probability density $\rho$. Associated with the random sample we have the closet point projection map $\sigma_n : X \to \{x_1, \ldots, x_n\} \subset X$ that satisfies

$$\|x - \sigma_n(x)\|_X = \min_{1 \leq i \leq n} \{\|x - x_i\|_X\}$$

for all $x \in X$. We recall that $W^{1,\infty}(X; Y)$ is the space of Lipschitz mappings from $X$ to $Y$. Throughout this section, $C, c > 0$ denote positive constants depending only on $\mathcal{M}$, and we assume $C \geq 1$ and $0 < c < 1$. We follow the analysis tradition of allowing the particular values of $C$ and $c$ to change from line to line.

We establish that that minimizers of (5) are unique on $\mathcal{M}$ in Theorem A.1, which follows from the strict convexity of the loss restricted to the data manifold $\mathcal{M}$. See also Figure 3 which shows how the solutions need not be unique off the data manifold.

Our first result is in the case where $\text{Lip}[u_0] \leq L_0$, and so the Lipschitz regularizer is not fully active. This corresponds to the case of clean labels. We state our result in generality, for approximate minimizers of (1), and specialize to the case $\text{Lip}[u_0] \leq L_0$ in Remark 2.8.

**Theorem 2.7** (Convergence result). *Assume* $\inf_{x \in \mathcal{M}} \rho(x) > 0$. *For any* $t > 0$, *with probability at least* $1 - Ct^{-1}n^{-(ct-1)}$ *every sequence* $u_n \in W^{1,\infty}(X;Y)$ *with zero empirical loss* $L[u_0, \rho_n] = 0$ *satisfies*

$$\|u_0 - u_n\|_{L^\infty(\mathcal{M};Y)} \leq C(L_0 + \text{Lip}[u_n]) \left( \frac{t \log(n)}{n} \right)^{1/m}.$$

*Remark* 2.8. If $u_n \in W^{1,\infty}(X;Y)$ is any sequence of minimizers of (1) and $\text{Lip}[u_0] \leq L_0$, then $J[u_n] \leq J[u_0] = 0$. Thus, $\text{Lip}[u_n] \leq L_0$ and Theorem 2.7 applies to the sequence $u_n$, yielding

$$\|u_0 - u_n\|_{L^\infty(\mathcal{M};Y)} \leq CL_0 \left( \frac{t \log(n)}{n} \right)^{1/m}.$$

It is important to note that Theorem 2.7 does not requires $u_n$ to be minimizers of (1)—we just require zero empirical loss, which is often achieved in practice (Zhang et al., 2016). This allows for approximation errors in solving (1) on the whole domain $X$, due to the restriction that $u$ must be expressed via a Deep Neural Network.

As an immediate corollary, we can prove that the generalization loss converges to zero, and so we obtain generalization.

**Corollary 2.9.** *Assume that for some* $q \geq 1$ *the loss* $\ell$ *satisfies*

$$\ell(y, y_0) \leq C\|y - y_0\|_Y^q \text{ for all } y_0, y \in Y. \tag{7}$$

*Then under the assumptions of Theorem 2.7*

$$L[u_n, \rho] \leq C(L_0 + \text{Lip}[u_n])^q \left( \frac{t \log(n)}{n} \right)^{q/m}$$

*holds with probability at least* $1 - Ct^{-1}n^{-(ct-1)}$.

*Proof.* By (7), we can bound the generalization loss as follows

$$L[u_n, \rho] = \int_{\mathcal{M}} \ell(u_n(x), u_0(x))\rho(x) \, dVol(x) \leq C\|u_n - u_0\|_{L^\infty(\mathcal{M};Y)}^q.$$

The proof is completed by invoking Theorem 2.7. $\qquad \square$

We now turn to the proof of Theorem 2.7, which requires a bound on the distance between the closest point projection $\sigma_n$ and the identity. The result is standard in probability, and we include it for completeness in Lemma 2.10 proved in §A.1. We refer the interested reader to (Penrose et al., 2003) for more details.

**Lemma 2.10.** *Suppose that* $\inf_{\mathcal{M}} \rho > 0$. *Then for any* $t > 0$

$$\|Id - \sigma_n\|_{L^\infty(\mathcal{M};X)} \leq C \left( \frac{t \log(n)}{n} \right)^{1/m}$$

*with probability at least* $1 - Ct^{-1}n^{-(ct-1)}$.

We now give the proof of Theorem 2.7.

*Proof of Theorem 2.7.* Since $L[u_n, \rho_n] = 0$ we have $u_0(x_i) = u_n(x_i)$ for all $1 \leq i \leq n$. Thus for any $x \in X$ we have

$$\|u_0(x) - u_n(x)\|_Y = \|u_0(x) - u_0(\sigma_n(x)) + u_0(\sigma_n(x)) - u_n(\sigma_n(x)) + u_n(\sigma_n(x)) - u_n(x)\|_Y$$

$$\leq \|u_0(x) - u_0(\sigma_n(x))\|_Y + \|u_n(\sigma_n(x)) - u_n(x)\|_Y$$
$$\leq (L_0 + \text{Lip}[u_n])\|x - \sigma_n(x)\|_X.$$

Therefore, we deduce

$$\|u_0 - u_n\|_{L^\infty(\mathcal{M};Y)} \leq (L_0 + \text{Lip}[u_n])\|Id - \sigma_n\|_{L^\infty(\mathcal{M};X)}.$$

The proof is completed by invoking Lemma 2.10. $\qquad \square$

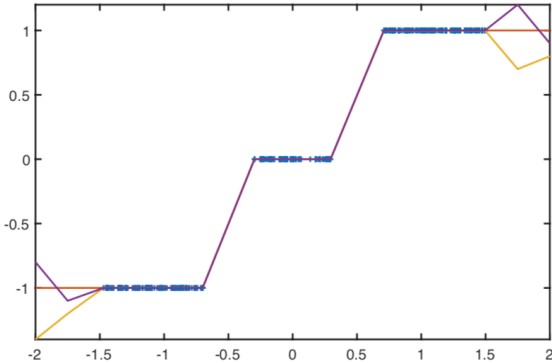

Figure 3: On the data manifold there is only one minimizer. Off the data manifold, there can be multiple minimizers.

## 2.4 CONVERGENCE FOR DIRTY LABELS

We now consider the setting of Problem 1.1, illustrated in Figure 1 right. We assume that we only have access to a "dirty" label function, which corresponds to an additive error of the form

$$u_0 = u_{\text{clean}} + u_e$$

where $u_{\text{clean}}$ is the label function, and $u_e : X \to Y$ is some error function, which is assumed to be zero with high probability. Assume that the error vector $e$ has a much larger Lipschitz constant than the labels, so that $\text{Lip}(u_0) \gg \text{Lip}(u_{\text{clean}})$.

We wish to fit the clean labels, while not fitting the errors, having access only to $u_0$. The labels correspond to the subset of the data which generate the low Lipschitz constant $L_{\text{clean}}$, while the errors correspond to pairs of labels that generate a high Lipschitz constant. Thus $L_{\text{clean}}$ can easily be estimated from the distribution of the pairwise Lipschitz constants of the data. With the goal in mind, we set $L_0 = L_{\text{clean}}$ in (1). The Lipschitz regularizer is active in (1), which can lead to the solution succeeding in avoiding the dirty labels, as in Figure 1 right.

Our main results (Theorems 2.12 and 2.11) show that minimizers of $J^n$ converge to minimizers of $J$ almost surely as the number of training points $n$ tends to $\infty$. It is beyond the scope of this work to estimate to what extent the errors are corrected, however we do know that the solution cannot fit $u_0$ due to the value of the Lipschitz constant, which is already an improvement over the case $\lambda = 0$.

The proofs for this section can be found in Section A.2.

**Theorem 2.11.** *Suppose that $\ell : Y \times Y \to \mathbb{R}$ is Lipschitz and strongly convex and let $L = \text{Lip}(u_0)$. Then for any $t > 0$, with probability at least $1 - 2t^{-\frac{m}{m+2}} n^{-(ct-1)}$ all minimizing sequences $u_n$ of (1) and all minimizers $u^*$ of (5) satisfy*

$$\frac{\theta}{2} \int_{\mathcal{M}} \|u_n - u^*\|_Y^2 \rho \, dVol(x) \le CL \left( \frac{t \log(n)}{n} \right)^{\frac{1}{m+2}}.$$

The next result drops the assumption of strong convexity of the loss.

**Theorem 2.12.** *Suppose that $\inf_{\mathcal{M}} \rho > 0$, $\ell : Y \times Y \to \mathbb{R}$ is Lipschitz, and let $u^* \in W^{1,\infty}(X; Y)$ be any minimizer of (5). Then with probability one*

$$u_n \longrightarrow u^* \quad \text{uniformly on } \mathcal{M} \text{ as } n \to \infty, \tag{8}$$

*where $u_n$ is any sequence of minimizers of (1). Furthermore, every uniformly convergent subsequence of $u_n$ converges on $X$ to a minimizer of (5).*

*Remark* 2.13. In Theorem 2.12 and Theorem 2.11, the sequence $u_n$ does not, in general, converge on the whole domain $X$. The important point is that the sequence converges on the data manifold $\mathcal{M}$, and solves the variational problem (5) off of the manifold, which ensures that the output of the DNN is stable with respect to the input. See Figure 3.

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

## A  PROOFS

### A.1  PROOFS FOR CLEAN LABELS

In this section we provide the proof of results stated in §2.3.

**Theorem A.1.** *Suppose the loss function satisfies Assumption 2.3. If $u, v \in W^{1,\infty}(X; Y)$ are two minimizers of (5) and $\inf_{\mathcal{M}} \rho > 0$ then $u = v$ on $\mathcal{M}$.*

*Proof.* Let $w = (u + v)/2$. Then

$$
\begin{aligned}
J[w] &= \int_{\mathcal{M}} \ell\left(\tfrac{1}{2}u + \tfrac{1}{2}v, u_0\right) \rho \, dVol(x) + \lambda \max\left(\mathrm{Lip}\left(\tfrac{1}{2}u + \tfrac{1}{2}v\right), 0\right) \\
&\leq \int_{\mathcal{M}} \left[\tfrac{1}{2}\ell\left(u, u_0\right) + \tfrac{1}{2}\ell\left(v, u_0\right)\right] \rho \, dVol(x) + \lambda \max\left(\tfrac{1}{2}\mathrm{Lip}\left(u\right) + \tfrac{1}{2}\mathrm{Lip}\left(v\right), 0\right) \\
&\leq \int_{\mathcal{M}} \left[\tfrac{1}{2}\ell\left(u, u_0\right) + \tfrac{1}{2}\ell\left(v, u_0\right)\right] \rho \, dVol(x) + \lambda \left[\tfrac{1}{2}\max\left(\mathrm{Lip}\left(u\right), 0\right) + \tfrac{1}{2}\max\left(\mathrm{Lip}\left(v\right), 0\right)\right] \\
&= \tfrac{1}{2}J[u] + \tfrac{1}{2}J[v] = \min_u J[u].
\end{aligned}
$$

Therefore, $w$ is a minimizer of $J$ and so we have equality above, which yields

$$
\int_{\mathcal{M}} \left[\tfrac{1}{2}\ell\left(u, u_0\right) + \tfrac{1}{2}\ell\left(v, u_0\right)\right] \rho \, dVol(x) = \int_{\mathcal{M}} \ell\left(\tfrac{1}{2}u + \tfrac{1}{2}v, u_0\right) \rho \, dVol(x).
$$

Since $\ell$ is strictly convex in its first argument, it follows that $u = v$ on $\mathcal{M}$. $\square$

*Proof of Lemma 2.10 of §2.3.* There exists $\epsilon_{\mathcal{M}}$ such that for any $0 < \epsilon \leq \epsilon_{\mathcal{M}}$, we can cover $\mathcal{M}$ with $N$ geodesic balls $B_1, B_2, \ldots, B_N$ of radius $\epsilon$, where $N \leq C\epsilon^{-m}$ and $C$ depends only on $\mathcal{M}$ (Györfi et al., 2006). Let $Z_i$ denote the number of random variables $x_1, \ldots, x_n$ falling in $B_i$. Then $Z_i \sim B(n, p_i)$, where $p_i = \int_{B_i} \rho(x) \, dVol(x)$. Since $\rho \geq \theta > 0$ and $Vol(B_i) \geq c\epsilon^m$ we have $p_i \geq c\epsilon^m$. Let $A_n$ denote the event that at least one $B_i$ is empty (i.e., $Z_i = 0$ for some $i$). Then by the union bound we deduce

$$
\begin{aligned}
\mathbb{P}(A_n) &\leq \sum_{i=1}^{N} \mathbb{P}\left(Z_i = 0\right) \\
&\leq C\epsilon^{-d}(1 - c\epsilon^m)^n \\
&= C\exp\left(n\log(1 - c\epsilon^m) - \log(\epsilon^m)\right) \\
&\leq C\exp\left(-cn\epsilon^m - \log(\epsilon^m)\right).
\end{aligned}
$$

Choose $0 < \epsilon \leq \epsilon_{\mathcal{M}}$ in the form $n\epsilon^m = t\log(n)$ with $t \leq n\epsilon_{\mathcal{M}}^m / \log(n)$. Then

$$
\mathbb{P}(A_n) \leq Ct^{-1}\exp\left(-(ct - 1)\log(n)\right).
$$

In the event that $A_n$ does not occur, then each $B_i$ has at least one point, and so $|x - \sigma_n(x)| \leq C\epsilon$ for all $x \in \mathcal{M}$. Therefore

$$\|\mathrm{Id} - \sigma_n\|_{L^\infty(\mathcal{M};X)} \leq C\epsilon = C \left( \frac{t \log(n)}{n} \right)^{1/m}$$

with probability at least $1 - Ct^{-1} \exp\left(-(ct - 1)\log(n)\right)$. Since $\|\mathrm{Id} - \sigma_n\|_{L^\infty(\mathcal{M};X)} \leq C\sqrt{d}$, the result holds for $t \geq n\epsilon_{\mathcal{M}}^m / \log(n)$, albeit with a larger constant $C$. $\qquad\square$

## A.2 Proofs for dirty labels

Here, we give the proofs of results from Section 2.4.

**Definition A.2.** We say that $\ell$ is strongly convex with parameter $\theta > 0$ if

$$\ell(ty_1 + (1 - t)y_2, y_0) + \tfrac{\theta}{2}t(1 - t)\|y_1 - y_2\|_Y^2 \leq t\ell(y_1, y_0) + (1 - t)\ell(y_2, y_0) \qquad (9)$$

for all $y_0, y_1, y_2 \in Y$ and $0 \leq t \leq 1$.

We note that when $\ell$ is twice differentiable, this notion of strong convexity is equivalent to assuming $\nabla_{y_1}^2 \ell \geq \theta I$. The definition in equation (9) is useful for non-smooth functions, such as the Lipschitz semi-norm present in $J[u]$.

We give a proposition useful in the proof of Lemma A.4.

**Proposition A.3.** *If $\ell$ is strongly convex with parameter $\theta > 0$ then*

$$J[tu_1 + (1 - t)u_2] + \tfrac{\theta}{2}t(1 - t) \int_{\mathcal{M}} \|u_1 - u_2\|_Y^2 \rho \, dVol(x) \leq tJ[u_1] + (1 - t)J[u_2]$$

*for all $u_1, u_2 \in W^{1,\infty}(X; Y)$ and $0 \leq t \leq 1$.*

*Proof.* We compute

$$\begin{aligned}
&J[tu_1 + (1 - t)u_2] \\
&= \int_{\mathcal{M}} \ell(tu_1 + (1 - t)u_2, u_0)\rho \, dVol(x) + \lambda \max\left(\mathrm{Lip}(tu_1 + (1 - t)u_2), 0\right) \\
&\leq tJ[u_1] + (1 - t)J[u_2] - \frac{\theta}{2}t(1 - t) \int_{\mathcal{M}} \|u_1 - u_2\|_Y^2 \rho \, dVol(x),
\end{aligned}$$

which completes the proof. $\qquad\square$

Before proving Theorem 2.11, we require a preliminary lemma.

**Lemma A.4.** *If $u^* \in W^{1,\infty}(X; Y)$ is a minimizer of (5) and $u \in W^{1,\infty}(X; Y)$ then*

$$\frac{\theta}{2} \int_{\mathcal{M}} \|u - u^*\|_Y^2 \rho \, dVol(x) \leq J[u] - J[u^*].$$

*Proof.* We use Proposition A.3 with $u_1 = u^*$ and $u_2 = u$ to obtain

$$J[tu^* + (1 - t)u] + \frac{\theta}{2}t(1 - t) \int_{\mathcal{M}} \|u^* - u\|_Y^2 \rho \, dVol(x) \leq tJ[u^*] + (1 - t)J[u].$$

Since $J[tu^* + (1 - t)u] \geq J[u^*]$

$$J[u^*] + \frac{\theta}{2}t(1 - t) \int_{\mathcal{M}} \|u^* - u\|_Y^2 \rho \, dVol(x) \leq tJ[u^*] + (1 - t)J[u],$$

and so

$$\frac{\theta}{2}t \int_{\mathcal{M}} \|u^* - u\|_Y^2 \rho \, dVol(x) \leq J[u] - J[u^*].$$

Setting $t = 1$ completes the proof. $\qquad\square$

The proof of Theorem 2.12 requires a preliminary Lemma. Let $H_L(X;Y)$ denote the collection of $L$-Lipschitz functions $w : X \to Y$.

**Lemma A.5.** *Suppose that $\inf_{\mathcal{M}} \rho > 0$, and $dim(\mathcal{M}) = m$. Then for any $t > 0$*

$$\sup_{w \in H_L(X;Y)} \left| \frac{1}{n} \sum_{i=1}^{n} w(x_i) - \int_{\mathcal{M}} w\rho \, dVol(x) \right| \leq CL \left( \frac{t \log(n)}{n} \right)^{\frac{1}{m+2}} \tag{10}$$

*holds with probability at least $1 - 2t^{-\frac{m}{m+2}} n^{-(ct-1)}$.*

The estimate (10) is called a discrepancy result (Talagrand, 2006; Györfi et al., 2006), and is a uniform version of concentration inequalities.

A key tool in the proof of Lemma A.5 is Bernstein's inequality (Boucheron et al., 2013), which we recall now for the reader's convenience. For $X_1, \ldots, X_n$ *i.i.d.* with variance $\sigma^2 = \mathbb{E}[(X_i - \mathbb{E}[X_i])^2]$, if $|X_i| \leq M$ almost surely for all $i$ then Bernstein's inequality states that for any $\epsilon > 0$

$$\mathbb{P}\left( \left| \frac{1}{n} \sum_{i=1}^{n} X_i - \mathbb{E}[X_i] \right| > \epsilon \right) \leq 2 \exp\left( -\frac{n\epsilon^2}{2\sigma^2 + 4M\epsilon/3} \right).$$

*Proof of Lemma A.5.* We note that it is sufficient to prove the result for $w \in H_L(X;Y)$ with $\int_{\mathcal{M}} w\rho \, dVol(x) = 0$. In this case, we have $w(x) = 0$ for some $x \in \mathcal{M}$, and so $\|w\|_{L^\infty(X;Y)} \leq CL$.

We first give the proof for $\mathcal{M} = X = [0,1]^m$. We partition $X$ into hypercubes $B_1, \ldots, B_N$ of side length $h > 0$, where $N = h^{-m}$. Let $Z_j$ denote the number of $x_1, \ldots, x_n$ falling in $B_j$. Then $Z_j$ is a Binomial random variable with parameters $n$ and $p_j = \int_{B_j} \rho \, dx \geq ch^m$. By the Bernstein inequality we have for each $j$ that

$$\mathbb{P}\left( \left| \frac{1}{n} Z_j - \int_{B_j} \rho \, dx \right| > \epsilon \right) \leq 2 \exp\left( -cnh^{-m}\epsilon^2 \right) \tag{11}$$

provided $0 < \epsilon \leq h^m$. Therefore, we deduce

$$\frac{1}{n} \sum_{i=1}^{n} w(x_i) \leq \frac{1}{n} \sum_{j=1}^{N} Z_j \max_{B_j} w$$

$$\overset{(11)}{\leq} \sum_{j=1}^{N} \left( \int_{B_j} \rho \, dx + \epsilon \right) \max_{B_j} w$$

$$\leq \sum_{j=1}^{N} \max_{B_j} w \int_{B_j} \rho \, dx + CLh^{-m}\epsilon$$

$$\leq \sum_{j=1}^{N} (\min_{B_j} w + CLh) \int_{B_j} \rho \, dx + CLh^{-m}\epsilon$$

$$\leq \sum_{j=1}^{N} \int_{B_j} w\rho \, dx + CLh^{-m}(h^{m+1} + \epsilon)$$

$$= \int_{X} w\rho \, dx + CL(h + h^{-m}\epsilon)$$

holds with probability at least $1 - 2h^{-m} \exp\left( -cnh^{-m}\epsilon^2 \right)$ for any $0 < \epsilon \leq h^m$. Choosing $\epsilon = h^{m+1}$ we have that

$$\left| \frac{1}{n} \sum_{i=1}^{n} w(x_i) - \int_{X} w\rho \, dx \right| \leq CLh$$

holds for all $u \in H_L(X;Y)$ with probability at least $1 - 2h^{-m} \exp\left( -cnh^{m+2} \right)$, provided $h \leq 1$. By selecting $nh^{m+2} = t\log(n)$

$$\sup_{w \in H_L(X;Y)} \left| \frac{1}{n} \sum_{i=1}^{n} w(x_i) - \int_{\mathcal{M}} w\rho \, dVol(x) \right| \leq CL \left( \frac{t \log(n)}{n} \right)^{\frac{1}{m+2}}$$

holds with probability at least $1 - 2t^{-\frac{m}{m+2}} n^{-(ct-1)}$ for $t \leq n/\log(n)$. Since we have $\|w\|_{L^\infty(X;Y)} \leq CL$, the estimate

$$\sup_{w \in H_L(X;Y)} \left| \frac{1}{n} \sum_{i=1}^{n} w(x_i) - \int_{\mathcal{M}} w\rho \, dVol(x) \right| \leq CL,$$

trivially holds, and hence we can allow $t > n/\log(n)$ as well.

We sketch here how to prove the result on the manifold $\mathcal{M}$. We cover $\mathcal{M}$ with $k$ geodesic balls of radius $\epsilon > 0$, denoted $B_{\mathcal{M}}(x_1, \epsilon), \ldots, B_{\mathcal{M}}(x_k, \epsilon)$, and let $\varphi_1, \ldots, \varphi_k$ be a partition of unity subordinate to this open covering of $\mathcal{M}$. For $\epsilon > 0$ sufficiently small, the Riemannian exponential map $\exp_x : B(0, \epsilon) \subset T_x\mathcal{M} \to \mathcal{M}$ is a diffeomorphism between the ball $B(0, r) \subset T_x\mathcal{M}$ and the geodesic ball $B_{\mathcal{M}}(x, \epsilon) \subset \mathcal{M}$, where $T_x\mathcal{M} \cong \mathbb{R}^m$. Furthermore, the Jacobian of $\exp_x$ at $v \in B(0, r) \subset T_x\mathcal{M}$, denoted by $J_x(v)$, satisfies (by the Rauch Comparison Theorem)

$$(1 + C|v|^2)^{-1} \leq J_x(v) \leq 1 + C|v|^2.$$

Therefore, we can run the argument above on the ball $B(0, r) \subset \mathbb{R}^m$ in the tangent space, lift the result to the geodesic ball $B_{\mathcal{M}}(x_i, \epsilon)$ via the Riemannian exponential map $\exp_x$, and apply the bound

$$\left| \frac{1}{n} \sum_{i=1}^{n} w(x_i) - \int_{\mathcal{M}} w\rho \, dVol(x) \right| \leq \sum_{j=1}^{k} \left| \frac{1}{n} \sum_{i=1}^{n} \varphi_j(x_i) w(x_i) - \int_{\mathcal{M}} \varphi_j w\rho \, dVol(x) \right|$$

to complete the proof. $\qquad\square$

*Remark* A.6. The exponent $1/(m+2)$ is not optimal, but affords a very simple proof. It is possible to prove a similar result with the optimal exponent $1/m$ in dimension $m \geq 3$, but the proof is significantly more involved. We refer the reader to (Talagrand, 2006) for details.

*Remark* A.7. The proof of Theorem 2.12 shows that (1) $\Gamma$-converges to (5) almost surely as $n \to \infty$ in the $L^\infty(X;Y)$ topology. $\Gamma$-convergence is a notion of convergence for functionals that ensures minimizers along a sequence of functionals converge to a minimizer of the $\Gamma$-limit. While we do not use the language of $\Gamma$-convergence here, the ideas are present in the proof of Theorem 2.12. We refer to (Braides, 2002) for details on $\Gamma$-convergence.

*Proof of Theorem 2.12.* By Lemma A.5 the event that

$$\lim_{n \to \infty} \sup_{w \in H_L(X;Y)} |L[w, \rho_n] - L[w, \rho]| = 0 \tag{12}$$

for all Lipschitz constants $L > 0$ has probability one. For the rest of the proof we restrict ourselves to this event.

Let $u_n \in W^{1,\infty}(X;Y)$ be a sequence of minimizers of (1), and let $u^* \in W^{1,\infty}(X;Y)$ be any minimizer of (5). Then since

$$\lambda(\mathrm{Lip}(u_n) - L_0) \leq J^n[u_n] \leq J^n[u_0] = \lambda(\mathrm{Lip}(u_0) - L_0)$$

we have $\mathrm{Lip}(u_n) \leq \mathrm{Lip}(u_0) =: L$ for all $n$. By the Arzelà-Ascoli Theorem (Rudin, 1976) there exists a subsequence $u_{n_j}$ and a function $u \in W^{1,\infty}(X;Y)$ such that $u_{n_j} \to u$ uniformly as $n_j \to \infty$. Note we also have $\mathrm{Lip}(u) \leq \liminf_{j \to \infty} \mathrm{Lip}(u_{n_j})$. Since

$$|L[u_n, \rho_n] - L[u, \rho]| \leq |L[u_n, \rho_n] - L[u, \rho_n]| + |L[u, \rho_n] - L[u, \rho]|$$
$$\leq C\|u_n - u\|_{L^\infty(\mathcal{M};Y)} + \sup_{w \in H_L(X;Y)} |L[w, \rho_n] - L[w, \rho]|$$

it follows from (12) that $L[u_{n_j}, \rho_{n_j}] \to L[u, \rho]$ as $j \to \infty$. It also follows from (12) that $J^n[u^*] \to J[u^*]$ as $n \to \infty$. Therefore

$$\begin{aligned}
J[u^*] &= \lim_{n \to \infty} J^n[u^*] \\
&\geq \liminf_{n \to \infty} J^n[u_n] \\
&= \liminf_{n \to \infty} L[u_n, \rho_n] + \lambda \max(\mathrm{Lip}(u_n) - L_0, 0) \\
&= \lim_{n \to \infty} L[u_n, \rho_n] + \liminf_{n \to \infty} \lambda \max(\mathrm{Lip}(u_n) - L_0, 0) \\
&\geq L[u, \rho] + \lambda \max(\mathrm{Lip}(u) - L_0, 0) = J[u].
\end{aligned}$$

Therefore, $u$ is a minimizer of $J$. By Theorem A.1, $u = u^*$ on $\mathcal{M}$, and so $u_{n_j} \to u^*$ uniformly on $\mathcal{M}$ as $j \to \infty$.

Now, suppose that (8) does not hold. Then there exists a subsequence $u_{n_j}$ and $\delta > 0$ such that

$$\max_{x \in \mathcal{M}} |u_{n_j}(x) - u^*(x)| > \delta$$

for all $j \geq 1$. However, we can apply the argument above to extract a further subsequence of $u_{n_j}$ that converges uniformly on $\mathcal{M}$ to $u^*$, which is a contradiction. This completes the proof. $\square$

*Proof of Theorem 2.11.* Let $L = \mathrm{Lip}(u_0)$. By Lemma A.5

$$\sup_{w \in H_L(X;Y)} |L[w, \rho_n] - L[w, \rho]| \leq CL \left( \frac{t \log(n)}{n} \right)^{\frac{1}{m+2}} \tag{13}$$

holds with probability at least $1 - 2t^{-\frac{m}{m+2}} n^{-(ct-1)}$ for any $t > 0$. Let us assume for the rest of the proof that (13) holds.

As in the proof of Theorem 2.12, we have $\mathrm{Lip}(u_n) \leq L$ and $\mathrm{Lip}(u^*) \leq L$, and so

$$|J^n[u^*] - J[u^*]|, |J^n[u_n] - J[u_n]| \leq CL \left( \frac{t \log(n)}{n} \right)^{\frac{1}{m+2}}.$$

Therefore

$$J[u_n] - J[u^*] = J^n[u_n] - J[u^*] + J[u_n] - J^n[u_n] \leq CL \left( \frac{t \log(n)}{n} \right)^{\frac{1}{m+2}}.$$

By Lemma A.4 we deduce

$$\frac{\theta}{2} \int_{\mathcal{M}} \|u_n - u^*\|_Y^2 \rho \, dVol(x) \leq CL \left( \frac{t \log(n)}{n} \right)^{\frac{1}{m+2}},$$

which completes the proof. $\square$

# B  INDUCED MATRIX NORMS

In some cases, we can take advantage of explicit formulas for matrix norms, which makes the estimates in (2) an explicit function of the weights. Define the induced matrix norm by

$$\|M\|_{p,q} = \sup_x \frac{\|Mx\|_q}{\|x\|_p}$$

Then the following matrix norms formulas hold (see (Horn et al., 1990, Chapter 5.6.4))

$$\|M\|_{\infty,\infty} = \max_i \sum_j |m_{ij}|, \qquad \|M\|_{1,1} = \max_j \sum_i |m_{ij}|$$

$$\|M\|_{1,\infty} = \max_{i,j} |m_{ij}|, \quad \|M\|_{2,\infty} = \max_i \sqrt{\sum_j m_{ij}^2}$$

# C  VARIATIONAL PROBLEMS IN IMAGE PROCESSING AND LIPSCHITZ EXTENSIONS

The variational problem (1) can be interpreted as a relaxation of the Lipschitz Extension problem.

$$\begin{cases} \min_{u:X \to Y} \mathrm{Lip}[u] \\ \text{subject to } u(x) = u_0(x) \text{ for } x \in \mathcal{D} \end{cases} \tag{LE}$$

for $\mathcal{D} \subset X$. The problem (LE) has more that one solution. Two classical results giving explicit solutions in one dimension go back to Kirzbaum and to McShane (McShane, 1934). However solving

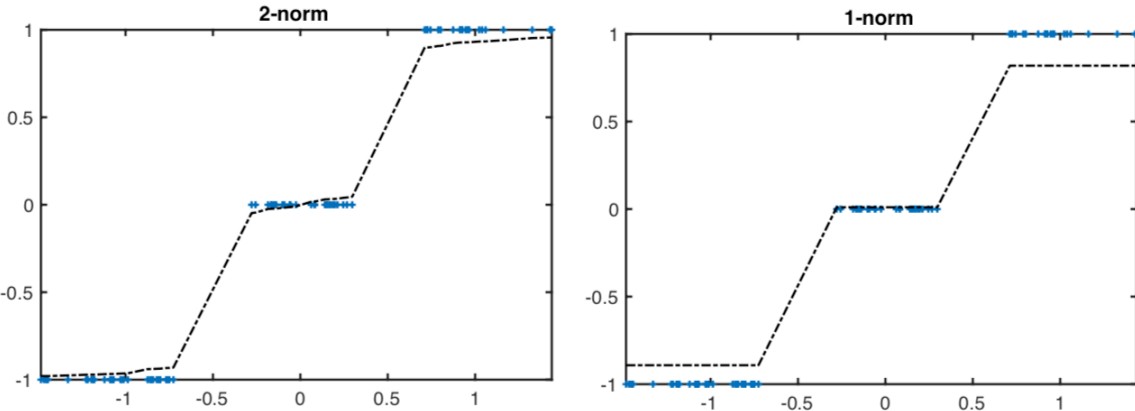

Figure 4: Comparison of different regularization methods. Lipschitz regularization preserves most of the labels (Figure 1). Tychonoff regularization smooths the solution (left). Total Variation regularization shifts the label values towards the mean (right).

(LE) is not practical for large scale problems. There has be extensive work on the Lipschitz Extension problem, see, (Johnson & Lindenstrauss, 1984), for example. More recently, optimal Lipschitz extensions have been studied, with connections to Partial Differential Equations, see (Aronsson et al., 2004). We can interpret (1) as a relaxed version of (LE), where $\lambda^{-1}$ is a parameter which replaces the unknown Lagrange multiplier for the constraint.

Variational problems are fundamental tools in mathematical approaches to image processing (Aubert & Kornprobst, 2006) and inverse problems more generally. Without regularization inverse problems can be ill-posed. The general form of the problem is

$$J[u] = L[u; u_0] + \lambda R[\nabla u] \qquad (14)$$

which combines a loss or *fidelity* functional, $L[u, u_0]$, which depends on the values of $u$ and the reference image $u_0$, and a *regularization* functional, $R[\nabla u]$, which depends on the gradient, $\nabla u$. The parameter $\lambda$ determines the relative strength of the two terms which emphasize fidelity versus regularization.

*Example* C.1. For example, a typical fidelity term is the standard least-squares $L[u, u_0] = \|u - u_0\|_{L^2(D)}^2$. The regularization $\||\nabla u(x)|\|_{L^2(D)}^2$ corresponds to the classical Tychonov regularization (Tikhonov & Arsenin, 1977), $R[\nabla u] = \||\nabla u(x)|\|_{L^1(D)}$ is the Total Variation regularization model of Rudin, Osher and Fatemi (Rudin et al., 1992).

Lipschitz regularization in not nearly as common. It appears in image processing in (Pock et al., 2010, §4.4) (Elion & Vese, 2007) and (Guillot & Le Guyader, 2009). Variational problems of the form (14) can be studied by the direct method in the calculus of variations (Dacorogna, 2007). The problem (14) can be discretized to obtain a finite dimensional convex convex optimization problem. The variational problem can also be studied by finding the first variation, which is a Partial Differential Equation (Evans, 1998), which can then be solved numerically. Both approaches are discussed in (Aubert & Kornprobst, 2006).

In Figure 4 we compare different regularization terms, in one dimension. The difference between the regularizers is more extreme in higher dimensions.

