# OpenReview forum: "Lipschitz regularized Deep Neural Networks generalize"
_ICLR.cc/2019/Conference_

### Official Review · AnonReviewer2 · 2018-10-22
**An interesting paper providing learning guarantees for unconstrained size neural network classifiers with explicit (exact) Lipschitz regularization.**

**Rating:** 7
**Confidence:** 4

**Review:**

The paper gives convergence guarantees to the true neural network classifier for the networks that are explicitly regularizing the (exact) Lipschitz constant of the network. The bound stated decays as ( log(n)/n )^{1/m}, where n is the number of training points and m is the dimension of the data manifold. Thus the decay rate is pretty slow when the data lies on a high-dimensional manifold. I believe it should also depend on the volume of the data manifold.

Computing the exact Lipschitz constant of the network is intractable. All the theorems apply to minimization problem defined in (1) with the exact Lip(u,X) being regularized, and not to the minimization problem of the lower bound (2). I believe there are also no convergence guarantees how quickly this lower bound on the Lipschitz constant approaches Lip(u,X), unless some assumptions are made on the smoothness of the data manifold (comments and insights would be appreciated). Thus in my opinion the current results have little practical importance. Nevertheless, it’s still interesting to see some ideal-setting guarantees being established.


Theorem 2.7: Is m == m_0 (the dimension of the data manifold)? Shouldn’t the bound be 2*C*L_0….? (assuming lemma 2.9 is correct)

Cor 2.8: Where does the volume Vol(M) of the manifold disappear? Is C in equation (6) the same C as in Theorem 2.7? Also, it looks like the bound should bound should have C^2 (assuming theorem 2.7 is correct, and the C’s are the same).


Introduction: I assume u(x,w) is not the last layer map, but a map from the input space to the labels (i.e., the whole neural network function and not just the map from the last hidden layer to the labels). If I am correct, it’s misleading to refer to u(x,w) as the last layer map. And if it is the last layer map, please justify why it is enough to consider the Lipschitz constant of the last layer.

The term “clean data” is never defined. My guess is that “clean data” refers to the realizable  setting, and “noisy” to agnostic, where the hypothesis space consist of neural networks of arbitrary size.


“Our analysis can avoid the dependence on the weights because we make the assumption that there are enough parameters so that u can perfectly fit the training data. The assumption is justified by Zhang et al. (2016).”
Note, that the network used in practice achieve zero classification error, as demonstrated by Zhang et al, but I doubt the cross entropy loss (that is usually being minimized) is exactly zero.

Remark 1.4 “will result in an expected loss..”  (there is also a typo here) should specify that you are talking about empirical error (0-1 loss), since I don’t think the loss function is fixed anywhere earlier in the text.

Remark 2.2 : Just wanted to note, that it is more common to call L(u,\rho) risk. The gap between L(u,\rho) and the empirical risk L(u,\rho_n) is usually called the generalization error (and only in the case of zero empirical risk, L(u,\rho) is equal to the generalization error). I did check the reference in Goodfellow et al. book, and I see that it is consistent with your definition.

Just below Remark 2.2:
“We would also expect the sequence of generalization losses L[u_n ; \rho] to converge to zero in the case of perfect generalization.”
Once again, this is true only in the realizable setting.

Could the authors comment on the connection to Cranko et al. 2018 work?

Typos:
In the abstract, no which: “..corrupted labels which we prove convergence to...”
“...a candidate measure for the Rademacher complexity, which a measure of generalization...”.
“1-Lipschitz networks with are also important for Wasserstein-GANs”
Section 2.1 “is it clear that u0, is a solution of “, should be “it is”


---------
[UPDATE]

Regarding the comment "Our paper resolves the question posed in ICLR Best paper 2017 "Understanding deep learning requires rethinking generalization"", I don't think that analyzing networks with explicit regularization resolves the questions stated in Zhang et al paper. As other reviewers mentioned, there are a number of other papers that formally define quantities that correlate with the generalization error, and are larger for random vs true labels. There are also other papers showing that one can tune some parameters of the optimization algorithm to avoid overfitting on random labels (while it is a modification to the algorithm, it is still similar to explicitly regularizing the Lipschitz constant of the network) (see e.g., Dziugaite et al work on SGLD).

Therefore, the claim in the abstract "A second result resolves a question posed in Zhang et al. (2016): how can a model distinguish between the case of clean labels, and randomized labels?" needs to be toned down a bit.

In my opinion, the work presented in this paper is a valuable contribution to learning theory. The new version of the paper is easy to read. Therefore, I recommend acceptance if the authors change the claim about resolving  the questions posed by Zhang et al.

Another typo:
 - for convergence, we require that the network also grow(s),

---

> ### Author Response · Authors · 2018-11-21
> **Reply to referee**
>
> We thank the referee for their comments. Below are our responses.
>
> Reviewer:-----
> Thus the decay rate is pretty slow when the data lies on a high-dimensional manifold...
>
> Reply:-----
> Yes, the constants in the theorems depend on various properties of the manifold, such as volume, curvature, etc.  The volume of the manifold comes into play via the covering arguments, and is absorbed into the constant C in Theorem 2.7. As we mentioned in the paper, we chose not to keep track of how the constant in the rate depends on the data manifold--we don’t know much about the data manifold in practice, and it does not affect the rate, just the constants.
>
> As explained in Remark 1.3 just before formula (2), the Lipschitz constant approximation described in (2) was implemented in (Finlay, 2018), and achieved significant improvement to state of the art adversarial robustness results, so the results do have practical importance.  There are many papers submitted to this conference which discuss approximating Lipschitz constants of Neural Networks.  Our approximation of the Lipschitz constant has a much smaller gap in the estimate than the ones which use the Bartlett results, and neglect the activation functions.
>
> Reviewer:-----
> Theorem 2.7: Is m == m_0?...
>
> Reply:-----
> Thanks for catching the m,m_0 typos. In the revision m=manifold dimension, D=number of labels. We follow the standard notation from analysis where C,c denote generic constants that can change from line to line. Here, 2C is renamed as C, so the factor of 2 is absorbed.  This can be confusing but it would be even messier to keep track of all the constants by name, and their exact values are unimportant. We added a remark to clarify this in the revision.
>
> Reviewer:-----
> Cor 2.8: Where does the volume Vol(M) of the manifold disappear?...
>
> Reply:-----
> As we mentioned above, the constants C in each theorem are slightly different, but are all given the same name, as is standard in analysis. The constants can depend on properties of the manifold M, such as volume and curvature, but do not depend on dimension. So in our notation, the volume vol(M) can be absorbed into the constants.
>
> Reviewer:-----
> I assume u(x,w) is not the last layer map...
>
> Reply:-----
> Yes, thanks for pointing this out, it was not precise. Clarified to say:
> "u(x,w) is the map from the input data to the last layer of the network."
>
> Reviewer:-----
> The term “clean data” is never defined...
>
> Reply:-----
> Thanks you for pointing this out. We corrected this to be more clear.
> In Section 2.3, which is now called “Convergence result”, the theorem applies to the label function u_0. In Section 2.4, we consider the case described in Problem 1.4: we are given a perturbed label function u_0 = u_true + e, where u_true has a Lipschitz constant of order 1, and the error (or noise) e has a Lipschitz constant much larger than 1. This is motivated by (Zhang, et al., 2016), who consider the case where a certain fraction of the data has incorrect labels.  For example, take two copies of x and add a small amount of noise to one of them, and then give it a different label.
>
> Reviewer:-----
> ...I doubt the cross entropy loss...is exactly zero...
>
> Reply:-----
> Technically, this is true that with cross entropy loss we may not train to zero loss.
> We included Theorem 2.7 because it given insight and a rate, in the case where the loss is zero. The case of non-zero loss is addressed by Theorems 2.12 and 2.15, which requires that we minimize the variational problem, but since the regularizer is active, we do not expect to have zero loss.  This is closer to common practice. However these theorems are mathematically heavier, so it’s worth including Theorem 2.7.
>
> Moreover, it would be simple to extend the results of Theorem 2.7 to the case where each u_n is nearly a minimizer, (say up to epsilon/n) and the theorem would still be true. There is always a trade-off between precision, generality, and clarity, and so we made a judgement call.
>
> Reviewer:-----
> Remark 1.4 typos
>
> Reply:-----
> Typo corrected and “expected loss" corrected to "expected empirical error"
>
>
> Reviewer:-----
> Just below Remark 2.2...this is true only in the realizable setting.
>
> Reply:-----
> We are indeed in the realizable setting.  However we removed this sentence, because it was redundant.
>
> Reviewer:-----
> Comment on the connection to Cranko et al. 2018 work?
>
> Reply:-----
> We assume you mean:  “Lipschitz Networks and Distributional Robustness” by Zac Cranko et al., 2018. (we were not aware of this paper, and we mention that our paper was posted on arxiv first).  This paper provides additional support for the merits of Lipschitz regularization by showing that it can arise as a special case of distributional robustness.
>
> Thanks for pointing it out.  We added a citation in Section 1.1 related work:
> (Cranko et al., 2018) show that Lipschitz regularization can arise as a special case of distributional robustness.
>
> Typos have been fixed.

---

> ### Author Response · Authors · 2018-11-28
> **Reply to updated comment about Zhang**
>
> The referee makes a good point that there are other responses to Zhang. The period for revisions ended on 11/26, so we are currently unable to revise the abstract. If there is an opportunity to revise the paper after acceptance, we would change the line in the abstract addressing Zhang to read
>
> "A second result is concerned with a question posed in Zhang et al. (2016): how can a model distinguish between the case of clean labels, and randomized labels?"
>
> instead of
>
> "A second result resolves a question posed in Zhang et al. (2016): how can a model distinguish between the case of clean labels, and randomized labels?"

---

### Official Review · AnonReviewer3 · 2018-11-02
**Potentially significant results, some question marks.**

**Rating:** 6
**Confidence:** 2

**Review:**

Disclaimer:  I am not a working expert in this specific area.  I have used spectral normalization for my own applications, and have working expertise in leveraging Lipschitz properties for various flavors of stability analyses.

This paper proposes a error convergence analysis for Lipschitz-regularized neural nets.  The analysis is framed in function space of the neural net and assumes the ability to solve the learning minimization problem.   The authors contrast their analysis with other analysis approaches in several ways.  First is that their analysis is "more direct" and second is that their analysis is independent of the learning approach (e.g., spectral normalization + SGD).


***Clarity***

The paper is mostly clearly written.  Some of the statements are presented without sufficient description.  For instance:

-- What does it mean when the paper states that their analysis is "more direct" than previous work?  There was no discussion of previous work beyond that comment.

-- The statement of Lemma 2.9 is not entirely clear.  Is \sigma_n a vector of white gaussian random variables?

-- Definition 2.3 precludes the hinge loss.  Comments?

-- Example 2.6, the regularized cross entropy loss doesn't satisfy L(y,z) = 0 if and only if y=z.  It might not even satisfy L >= 0.
 Comments?

-- Since the function is Lipschitz, in the noisy case, can one say anything about the guarantees near the manifold for some definition of near?  E.g., can one bound how bad the tail parts of Figure 3 can be?


***Originality***

It seems to me that the analysis aims to set up the problem such that one can leverage standard results in probability theory.    For instance, the proof of Theorem 2.7 is quite straightforward given Lemma 2.9.  Of course, setting up the problem properly is 90% of the work.  The analysis for the noisy case is much more involved.  It is unfortunately beyond my expertise to properly judge originality in this case.  Perhaps the authors can comment on how the way they set up the problem is novel?


***Significance***

My biggest question mark w.r.t. significance are the claims of how this analysis compares with previous work.

-- What does "more direct" analysis mean?

-- What is the significance of an algorithm-agnostic analysis?  I understand the appeal from a certain perspective, but can the authors point to previous literature (perhaps not in deep learning) where an algorithm-agnostic analysis was shown to give more insight?


***Overall Quality***

Conditioned on the problem setup being novel and the comparison with related work clarified, I think this is a solid contribution.

---

> ### Author Response · Authors · 2018-11-21
> **Reply to referee**
>
> We thank the referee for their comments. Below are our responses.
>
> Reviewer:----------
> Review: Disclaimer: I am not a working expert in this specific area. I have used spectral normalization for my own applications, and have working expertise in leveraging Lipschitz properties for various flavors of stability analyses.
>
> This paper proposes a error convergence analysis for Lipschitz-regularized neural nets. The analysis is framed in function space of the neural net and assumes the ability to solve the learning minimization problem. The authors contrast their analysis with other analysis approaches in several ways. First is that their analysis is "more direct" and second is that their analysis is independent of the learning approach (e.g., spectral normalization). The analysis for cl
>
> Reply:----------
> OK
>
> Reviewer:----------
> ***Clarity***
> The paper is mostly clearly written. Some of the statements are presented without sufficient description. For instance:
> -- What does it mean when the paper states that their analysis is "more direct" than previous work? There was no discussion of previous work beyond that comment.
>
> Reply:----------
> We removed that clause of the sentence, and kept “our analysis does not depend on the training method” which was the main point of the sentence.
>
> Reviewer:----------
> -- The statement of Lemma 2.9 is not entirely clear. Is \sigma_n a vector of white gaussian random variables?
>
> Reply:----------
> sigma_n was defined one paragraph above. It is the closest point projection.
>
> Reviewer:----------
> -- Definition 2.3 precludes the hinge loss. Comments?
>
> Reply:----------
> Indeed, we require strong convexity so the hinge loss is not allowed.  We discuss how the KL divergence is allowed, which is the more relevant loss
>
> Reviewer:----------
> -- Example 2.6, the regularized cross entropy loss doesn't satisfy L(y,z) = 0 if and only if y=z. It might not even satisfy L >= 0. Comments?
> Reply:----------
> The regularized cross entropy loss does satisfy these properties if you shift by the appropriate constant.  We added a comment to clarify this point.
>
> Reviewer:----------
> -- Since the function is Lipschitz, in the noisy case, can one say anything about the guarantees near the manifold for some definition of near? E.g., can one bound how bad the tail parts of Figure 3 can be?
>
> Reply:----------
> Off the manifold we still know that the function is Lipschitz. The tail parts are bounded by the Lipschitz constant of the function, so they cannot change rapidly as you move away from the manifold.
>
> Reviewer:----------
> ***Originality***
>
> It seems to me that the analysis aims to set up the problem such that one can leverage standard results in probability theory. For instance, the proof of Theorem 2.7 is quite straightforward given Lemma 2.9. Of course, setting up the problem properly is 90% of the work. The analysis for the noisy case is much more involved. It is unfortunately beyond my expertise to properly judge originality in this case. Perhaps the authors can comment on how the way they set up the problem is novel?
>
> Reply:----------
> Variational problems are quite common in Image Processing, for example, the Total Variation Regularization was developed in the 1990’s for image denoising by Stan Osher, and Sapiro developed the Inpainting Model for filling in missing parts of images.  As the referee points out, the probabilistic techniques we use are standard. However, the combination of variational models and probabilistic techniques is not as common, and the application of these techniques to regularization in deep learning is new. Section 5 in the appendix discusses the connection to these works.
>
> Based on your advice we added a sentence in Section 1.1. Related work, “our analysis has more in common with Total Variation denoising … see Appendix C”
>
> Reviewer:----------
> ***Significance***
>
> My biggest question mark w.r.t. significance are the claims of how this analysis compares with previous work.
>
> -- What does "more direct" analysis mean?
> -- What is the significance of an algorithm-agnostic analysis? I understand the appeal from a certain perspective, but can the authors point to previous literature (perhaps not in deep learning) where an algorithm-agnostic analysis was shown to give more insight?
>
> Reply:----------
> In practice, the algorithm matters for performing the minimization.  However, convergence means that, asymptotically in the limit n to infinity,  the quality of the results are independent of the algorithm used to do the minimization.  This is in contrast to works which claim that SGD leads to generalization.  For these works, the quality of results may depend on the initialization and hyperparameters (learning rate schedule) of SGD.

---

### Official Review · AnonReviewer1 · 2018-11-02
**interesting approach to generalization, but are the guarantees relevant?**

**Rating:** 4
**Confidence:** 3

**Review:**

The paper studies generalization through a general empirical risk minimization procedure with Lipschitz regularization.
Generalization is measured through distance of the empirical minimizer function to a true labeling function u_0, or to the minimizer of the expected regularized loss.

The approach of studying generalization through the lens of Lipschitz stability over the data is interesting,
and the study directly considers minimizers of regularized optimization objectives, which is different than
recent generalization bounds which often provide guarantees on a given network regardless of how it was trained.

However, various aspects of the setup seem quite disconnected from practice in the context of deep neural networks,
and the obtained guarantees are quite weak and not always connected to generalization:

- the approach relies on a constant L_0 in the objective which is assumed known in advance (although it is usually set to 0 in practical methods), and determines the nature of convergence. In particular, the results for small L_0 (referred to as "noisy labels") only shows convergence to a minimizer u^* of the expected *regularized* loss, thus does not characterize generalization in the usual sense since u^* is biased. More generally, the distinction between 'clean' and 'noisy' labels is confusing and should be clarified: the paper seems to assume that the true labeling function u_0 may itself produce incorrect labels deterministically even with infinite data, which is an odd way to formulate the learning problem.

- the convergence rates obtained in the paper exhibit a curse of dimensionality (O(n^{-1/m}) where m is the dimensionality of the data manifold). Given that most other bounds for neural networks do not exhibit such a dependence on dimension, this seems to be a weaker guarantee, unless the setting captures an improvement in a different setting. (edit: I removed the previous remark on parametric rates, which was inaccurate) Some of the constants also seem to grow exponentially with m. Either way, this should be discussed in the paper.

- possibly related to the previous point, all the theory in the paper is agnostic to the function class considered, given that it simply considers all lipschitz functions in the variational problem (1). Given that the authors attempt to explain generalization of neural networks, this seems like a non-negligible disconnect since there could be approximation errors. In particular, even if deep networks can perfectly fit training data, it is not clear that they can achieve the best trade-off with the Lipschitz constant in the regularized objective (1).

- the assumptions on the prediction space (simplex) and the used loss functions also seem disconnected from practice (the cross-entropy loss usually includes a softmax). Note that while usual networks can fit randomly labeled data (Zhang et al.), this does not mean their loss is 0. Yet the analysis seem to rely on having zero loss on training points, even in the case of 'noisy labels'.

Additionally, the use of covering arguments in the input space in the proposed approach is related to the study of generalization through robustness [Xu & Mannor (2010), "Robustness and Generalization"]. The authors should discuss the relationship to this work.

More comments:
- the term 'converge' in the title is not clear. In the abstract, what does 'verification' mean?
- Section 2.3:
  m_0 == m ?
  How does C grow with m in Theorem 2.7 and Corollary 2.8?
  What is meant by 'perfect generalization'?
  Is eq. (6) realistic for classification losses?
- Section 2.4: clarify what is meant by 'noisy labels'

---

> ### Author Response · Authors · 2018-11-21
> **Reply to referee**
>
> We thank the referee for their comments. Below are our responses.
>
> Reviewer:----------
> - the approach relies on a constant L_0...
>
> Reply:----------
> As we discussed in remark 1.4, what we have in mind is the following example, inspired by the work of (Zhang et al., 2016).  Suppose, for example, that 10 percent of the labels have been changed.  Now the histogram of the terms $\| y_i - y_j |/|x_i-x_j|$ has two modes: the clean data has Lipschitz constant of O(1), and the randomized data terms are  large.  Now we train with the smaller of the constants.  The convergence theorem says we will converge to a function different than the noise label function.  We obtain a desirable result: this function won’t learn the high Lipschitz constant labels, because it is constrained to have a smaller Lipschitz.  In fact we have experimented with detecting incorrect labels using this method.
>
> Reviewer:----------
> - the convergence rates exhibit a curse of dimensionality...
>
> Reply:----------
> We agree the rate suffers from the curse of dimensionality. This rate is optimal in the non-parametric regime, where the network can learn almost any function, and is simply based on sampling the manifold densely. We are not aware of any parametric rates for generalization of neural networks. Obtaining better parametric rates requires placing serious constraints on the class of regression or classification functions.
>
> Reviewer:----------
> -all the theory in the paper is agnostic to the function class considered...
>
> Reply:----------
> In the revision, the statement of Theorem 2.7 has been elaborated to emphasize that we only require zero empirical loss and a Lipschitz constant, so the theorem applies to approximate minimizers. We note that when we don’t train to zero loss (such as with regularizations used in common practice), Theorems 2.12 and 2.15 can be applied, which do not require zero empirical loss.
>
> Reviewer:----------
> - the assumptions on the prediction space/loss seem disconnected from practice...
>
> Reply:----------
> We allow for general prediction space and loss functions, and used the probability simplex as an example (Example 2.5).  For classification, we choose to group softmax with the final layer of the neural network, so that the output u(x) of the network lies in the probability simplex, and the loss is just cross-entropy without softmax. We expanded Example 2.6 in the revision to clarify. We point out that only the proof of Theorem 2.7 assumes zero empirical loss; Theorems 2.12 and 2.15 hold in general without zero loss, which is often the case when one includes regularization.
>
> Reviewer:----------
> Use of covering arguments related to [Xu & Mannor (2010)]...
>
> Reply:----------
> Covering arguments are standard arguments in probability and other fields. In the Euclidean setting the result can be found in graduate level probability textbooks. The manifold case is only slightly more general, and for that case we cited the textbook [Gyorfi et al., 2006]. We included the proof in the appendix for reference but made it clear that this is a standard result.
>
> Reviewer:----------
>  - the term 'converge' in the title is not clear...what does 'verification' mean?
>
> Reply:----------
> Convergence means that the sequence of functions u_n converges uniformly on X to the limiting function u, which is the label function, in Theorem 2.7, and the minimizer of the limiting function in the general case of Theorems 2.12 and 2.15. This is described in the paper. We did not mean anything specific by ‘verification’ in the abstract and have removed this in the revision.
>
>
> Reviewer:----------
> - the term 'converge' in the title is not clear. In the abstract, what does 'verification' mean?
> - Section 2.3:
>   m_0 == m ?
>   How does C grow with m in Theorem 2.7 and Corollary 2.8?
>   What is meant by 'perfect generalization'?
>   Is eq. (6) realistic for classification losses?
> - Section 2.4: clarify what is meant by 'noisy labels'
>
> Reply:----------
> We have fixed the m_0,m typos in the revision (m denotes the data manifold dimension, and D is the number of labels). The constant C is independent of the dimension m of the data manifold. We corrected 'perfect generalization' to 'generalization'.
>
> Yes, eq. (6) is realistic. In the revision, we expanded Example 2.6 to explain further. In the standard classification case where softmax is used, Eq (6) applies with q = 1 to cross-entropy loss provided the inputs to softmax are bounded. In the more general case, we explain in Example 2.6 how regularized cross-entropy can be used when softmax may report labels very close to zero.
>
> Please see the reply above referencing (Zhang et al., 2016) for clarification on noisy labels.

---

> > ### Comment · AnonReviewer1 · 2018-11-22
> > **still puzzled**
> >
> > I remain puzzled about the setting and goal considered, particularly for the noisy setting: the additive 'noise' function e() defined in Section 2.4 seems to be fixed in advance and deterministic, which is an odd way to formulate how 'noisy' labels are generated in the data distribution, given that there is no noise (maybe a better way to describe the experiments of Zhang et al. would be to have labels y_i = u_true(x_i) + e_i, where e_i is noise).
> > Also, the authors show convergence to u^*, and it is not clear if u^* is related to u_true at all. A reasonable learning algorithm would try to converge to u_true, or at least to obtain small excess risk with respect to it.
> >
> > Regarding the curse of dimensionality, I agree that this does not make the bound uninteresting, however, the authors should at least describe a scenario in which they think the obtained bound yields some insight. Note that in the case of neural networks, existing bounds have O(1/sqrt(n)) rates assuming large enough margins and low complexity (e.g. Bartlett et al.), so if the authors believe the assumptions in such bounds do not capture certain scenarios in practice, I encourage them to discuss why.
> >
> > Concerning Xu and Mannor, I was just referring to the fact that their results seem to leverage local robustness of the prediction function in a similar way to the present paper (the dependence on covering numbers just follows from this), hence I believe a comparison would be appropriate.

---

> > > ### Author Response · Authors · 2018-11-23
> > > **further clarification about the noisy setting**
> > >
> > > We made an effort in our revision to clarify the setting of Section 2.4 where the given label function may have errors.  These errors can be random, or fixed.  For example, in the data bootstrapping setting, the labels could come from another model, which has 5 percent error.
> > >
> > > The section, which is a response to the Zhang paper, is proof of principle which shows that our model can distinguish between a "clean" label function and a label function with errors.  The way this distinction is made is illustrated in Figure 1b: the data points which would increase the Lipschitz constant beyond a certain threshold are not learned.   The point of the model is that, in contrast to unregularized models,  it does not learn the "dirty" label function.  It is unrealistic to expect that the model will learn the true underlying label function, which is not given.   However, as the figure shows, after thresholding, it may learn the correct labels.  Work in progress is to establish conditions when this can happen, however it a much more technical result which is beyond the scope of this paper.

---

> > > ### Author Response · Authors · 2018-11-24
> > > **Comparison with Bartlett 1997 rates**
> > >
> > > Let us add a few comments about the O(1/sqrt(n)) rate quoted from (Bartlett, 1997).
> > >
> > > This rate is for missclassification error, so we are comparing apples to oranges. We bound the difference |u_n-u_0| between the true label function u_0 and the learned function u_n. In practice, u_n(x) is rounded to the nearest label, so once |u_n-u_0| < 1/2, all classification results will be correct after rounding. Furthermore, Bartlet's rate is actually O(A^{k(k+1)/2}sqrt(log(d))/sqrt(n)), where all weights in the network are assumed to be bounded by A, and k is the depth of the neural network. It is common in deep learning for k to be large (say k > 50), and so even if A=2, the prefactor is enormous. In other words, the rate is vacuous for networks that do not have very low complexity. In modern applications, the networks have much higher complexity, as was shown in (Zhang et al., 2016), and the high complexity is closely related to adversarial examples. It is interesting to point out that A^k is related to the very loose bound on the Lipschtiz constant of the network that we note in the paper (see Eq. (3)). It was shown in (Finlay, 2018) that this is an overestimate of the Lipschitz constant of the network by many orders of magnitude.
> > >
> > > We have made a minor revision to the paper with some additional comments about the Bartlett 1997 rate, and a citation to Xu and Mannor. There are many works connecting robustness (in some sense) with generalization (e.g., Cranko, et al. 2018 connect robustness to Lipschitz regularization, which is more relevant to our paper, and Bousquet et al., 2002).

---

### Author Response · Authors · 2018-11-21
**General remarks**

Thanks to the referee's for their comments. We have replied point by point to the referee comments below.

Here are some general replies to AnonReviewer1.

I.  The point of the statement of our theorem is that the Lipschitz constant of the trained network is a coefficient in the convergence rate, which suggests that better performance comes from training for a small Lipschitz constant. Indeed, in a separate paper (Finlay, 2018), which we cited, Lipschitz regularization was implemented and led to state of the art adversarial robustness. The referee is pointing out that the rate is vacuous when the manifold dimension m is large, which is true. However, we are unaware of any non-vacuous rates that apply to deep learning.


II.  Now consider the setting of (Zhang et al., 2016).  Suppose, for example, that 10 percent of the labels have been changed.  Now the histogram of the terms $\| y_i - y_j |/|x_i-x_j|$ has two modes: the clean data has Lipschitz constant of  bump  order 1, and the randomized data terms are  large.  Now we train with the smaller of the constants.  The convergence theorem says we will converge to a function different than the noise label function.  We obtain a desirable result: this function won’t learn the high Lipschitz constant labels, because it is constrained to have a smaller Lipschitz.  In fact we have experimented with detecting incorrect labels using this method.
Even though in practice, the softmax is often coded into the cross-entropy loss, for the purpose of the theory, we are considering the softmax to be the last layer of the network.  In addition, this allows for other novel methods of projecting onto the simplex to be used.

---

### Author Response · Authors · 2018-11-26
**Revision note: emphasized the Zhang connection, made the paper easier to read**

We reworked the paper to make it easier to read, moving longer proofs to the appendix, and clearly emphasizing the connection to the question posed by Zhang 2016 (see the abstract and second paragraph).    We cut the paper down to 7 pages, and fixed minor typos.

---

> ### Comment · AnonReviewer3 · 2018-12-13
> **much clearer**
>
> Thanks for the revision.  The key take-aways of the paper are much clearer to me now.  One minor comment is that the exposition of the introduction is a bit too tilted towards the Zhang connection. In particular, I think the first paragraph in the intro could be longer & more fleshed out.

---

> > ### Author Response · Authors · 2018-12-19
> > **We can do that**
> >
> > Thanks for the comment.  We can certainly expand the introduction.

---

### Meta-Review · Area_Chair1 · 2018-12-16
**Cannot escape the curse**

**Confidence:** 4
**Recommendation:** Reject

**Metareview:**

This paper is entitled Lipschitz regularized deep networks generalize. In fact, the paper has nothing in particular to do with neural networks. It is really the study of minimizers of a Lipschitz-regularized risk functional over certain nonparametric classes. The connection with neural networks is simply that one can usually achieve zero empirical risk for (overparametrized) neural networks and so, in deep learning practice, neural networks behave like a nonparametric class. Given the lack of connection with neural networks, one cannot logically learn anything specific about neural networks from this paper. It should be renamed... perhaps "Lipschitz regularization with an application to deep learning".  One could raise issues of technical novelty, as it seems many of the key results are known.

I also question the insight that the bounds provide: they end up depending exponentially on the dimension of the data manifold. In the noiseless case, this exponential dependence arises from a triangle inequality between an arbitrary data point x and the nearest training data point! In the noisy case, this exponential dependence appears in a nonasymptotic uniform law of large numbers over the class of L-Lipschitz functions. There's no insight into deep learning here. It's also hard to judge whether these rates are what is explaining deep learning practice: it's unclear what the manifold  dimensionality is, but it seems unlikely that this bound explains empirical performance (even if it describes the asymptotic rate of convergence).

One of the main results shows that, in the face of corrupted label (corrupted in a particular way), Lipschitz regularization can ```"undo" the corruption. However, convergence is not measured with respect to the true labeling function, but rather to the solution to the population regularized risk functional. How this solution relates to the true labeling function is unclear.

The paper also purports to resolve a mystery of generalization raised by Zhang et al (ICLR 2017). In that paper, the authors point to the diametrically opposed generalization performance on "true" and "random" labels. In fact, this paper does not resolve this problem because Zhang et al. were interested in how SGD solves this problem without explicit regularization. That Lipschitz regularization could solve this problem is borderline obvious.

I wanted to make a few comments.

In the rebuttal with reviewers, the question of parametric rates comes up. I think there's some confusion on both the part of the reviewer and authors. The parametric rates are often apparent but not real. The complexity terms often have an uncharacterized dependence on the number of data (through the learning algorithm) and on the size of the network (which is implicitly chosen based on the data complexity). In practice, these bounds are vacuous.

At some point, the authors argue that "In practice, u_n(x) is rounded to the nearest label, so once |u_n-u_0| < 1/2, all classification results will be correct after rounding." I'm not entirely sure I understand the logic here. First, convergence to u_0 is not controlled, but rather convergence to u*. u* may spend most of its time near the decision boundary, rendering uniform convergence almost useless. One would need noise conditions (Tysbakov) to make some claim.

Some other issues:

1. in (1), u ranges over X\to Y, but is then applied also to a weight vector.

2.  Is"continuum variational problem" jargon? If so, cite. Otherwise, taking limits of rho_n and J makes sense only if J is suitably continuous, which depends on the loss function. You later address this convergence and so you should foreshadow.

3. Notation L[u;\rho] in (5) should be L[u,\rho], no?

4. (Goodfellow et al., 2016, Section 5.2) is an inappropriate citation for the term "Generalization".

5. in Thm 2.7.,  there is reference to a sequence mu_n and i assume the sequence elements is indexed by n, but then  n appears in the probability with which the bound holds, and so this bound is not about the sequence but about a solution for \rho_n for fixed n.

6. Id should not be italicized in the statement of Lemma 2.10.  Use mathrm not text/textrm.  it should also be defined.

7. "convex convex" typo.